# Natural Products Lysobactin and Sorangicin A Show *In Vitro* Activity against *Mycobacterium abscessus* Complex

Jaryd R. Sullivan,[a,b,c] Jacqueline Yao,[b] Christophe Courtine,[a,b] Andréanne Lupien,[b,c] Jennifer Herrmann,[d] Rolf Müller,[d,e] Marcel A. Behr[a,b,c,f]

aDepartment of Microbiology & Immunology, McGill University, Montréal, Québec, Canada

bInfectious Diseases and Immunity in Global Health Program, Research Institute of the McGill University Health Centre, Montréal, Québec, Canada

cMcGill International TB Centre, Montréal, Québec, Canada

dDepartment of Microbial Natural Products, Helmholtz-Institute for Pharmaceutical Research Saarland (HIPS), Helmholtz Centre for Infection Research (HZI), Saarbrücken, Germany

eDepartment of Pharmacy, Saarland University, Saarbrücken, Germany

fDepartment of Medicine, McGill University Health Centre, Montréal, Québec, Canada

**ABSTRACT** The prevalence of lung disease caused by *Mycobacterium abscessus* is increasing among patients with cystic fibrosis. *M. abscessus* is a multidrug resistant opportunistic pathogen that is notoriously difficult to treat due to a lack of efficacious therapeutic regimens. Currently, there are no standard regimens, and treatment guidelines are based empirically on drug susceptibility testing. Thus, novel antibiotics are required. Natural products represent a vast pool of biologically active compounds that have a history of being a good source of antibiotics. Here, we screened a library of 517 natural products purified from fermentations of various bacteria, fungi, and plants against *M. abscessus* ATCC 19977. Lysobactin and sorangicin A were active against the *M. abscessus* complex and drug resistant clinical isolates. These natural products merit further consideration to be included in the *M. abscessus* drug pipeline.

**IMPORTANCE** The many thousands of people living with cystic fibrosis are at a greater risk of developing a chronic lung infection caused by *Mycobacterium abscessus*. Since *M. abscessus* is clinically resistant to most anti-TB drugs available, treatment options are limited to macrolides. Despite macrolide-based therapies, cure rates for *M. abscessus* lung infections are 50%. Using an in-house library of curated natural products, we identified lysobactin and sorangicin A as novel scaffolds for the future development of antimicrobials for patients with *M. abscessus* infections.

**KEYWORDS** *Mycobacterium abscessus*, natural products, lysobactin, sorangicin A

**M**ycobacterium abscessus (*M. abscessus*) complex consists of three subspecies (*M. abscessus* subsp. *abscessus*, *M. abscessus* subsp. *massiliense*, *M. abscessus* subsp. *bolletii*) that cause disease in immunocompromised and immunocompetent hosts (1–5). Although *M. abscessus* was first isolated from a knee abscess in 1953, the most common clinical presentation is pulmonary disease in patients with cystic fibrosis (CF), chronic lung disease, or undergoing lung transplantation and is becoming more frequent (6–8). In some instances, extrapulmonary disease can result from dissemination or surgical site infections that lead to skin and soft tissue infections (6, 9).

Clinical management of *M. abscessus* pulmonary disease is challenged by notoriously drug resistant phenotypes and a lack of effective antimicrobials. *M. abscessus* is intrinsically resistant to most antitubercular agents used to treat tuberculosis despite having a homolog of the target (10). The treatment for *M. abscessus* pulmonary disease in a patient with CF includes an intensive phase of an oral macrolide like clarithromycin or azithromycin, intravenous amikacin, and an additional intravenous agent like cefoxitin or imipenem (11). The

Address correspondence to Marcel A. Behr, marcel.behr@mcgill.ca.

The authors declare no conflict of interest.

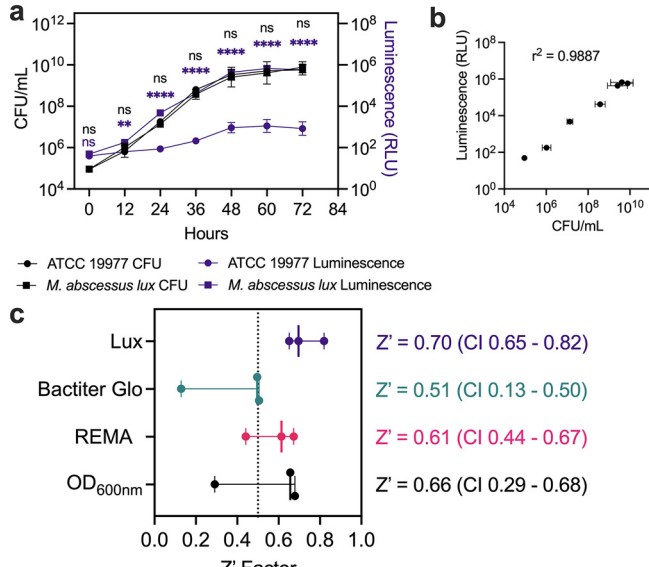

**FIG 1** Validation of luminescence from *M. abscessus* constitutively expressing *lux*. (a) *M. abscessus* 19977 ATCC 19977 was made to constitutively express the *luxCDABE* operon (*M. abscessus lux*). Reference strain ATCC 19977 (circles) and *M. abscessus lux* (squares) were grown in 7H9 complete with CFU/mL (black symbols) and luminence (purple symbols) measured at 12-h intervals. Data shown is $n = 3$ with mean $\pm$ SD. *P*-values from 2-way ANOVA with Sidak's multiple-comparison test. (b) Correlation between luminescence output and bacterial growth measured with Pearson's correlation coefficient. (c) Z' Factor measured using $OD_{600nm}$, resazurin microtiter assay (REMA), and Bactiter Glo readouts on *M. abscessus* ATCC 19977, and luminescence readout on *M. abscessus lux*. Data shown is median with 95% CI of $n = 3$.

intensive phase of treatment lasts for up to 3 months depending on the severity of the infection and the tolerability of the regimen. Subsequently, patients continue with the oral macrolide and substitute the injectable agents with oral clofazimine, minocycline, moxifloxacin, and nebulized amikacin for 14 months (11).

Treatment regimens can be further complicated by various susceptibilities among the subspecies where *M. abscessus* subsp. *abscessus* and *bolletii* display inducible resistance to macrolides conferred by the ribosomal methyltransferase encoded by *erm*(41), resulting in worse treatment outcomes. Comparatively, *M. abscessus* subsp. *massiliense* has a truncated *erm*(41) and higher treatment success rates (12–14). Consequently, the intrinsic resistance mechanisms of the bacteria and the lack of effective antimicrobials against *M. abscessus* necessitate additional drug discovery efforts to identify novel antimicrobials.

Natural products are a reservoir of genetically encoded microbial metabolites with vast chemical diversity (15, 16). These metabolites have been crafted by evolution to mediate chemical signaling roles and thus, possess the required properties for microbial penetration and an affinity for biological targets (17, 18). Considering that two-thirds of antibiotics used in the clinic are natural products or derived from a natural product scaffold, they are a proven source for antimicrobial discovery (16, 19). Here, we screened an in-house HZI/HIPS library of 517 natural products purified from fermentations of various bacteria (mostly myxobacteria) and fungi against *M. abscessus* ATCC 19977. We discovered that the cyclic depsipeptide lysobactin (LYB) and the macrolide sorangicin A (SOR) have activity against the reference strain and against a panel of drug resistant clinical isolates from CF patients.

## RESULTS

**Validation of *M. abscessus lux* for phenotypic screening.** To identify new antimicrobial agents against the difficult-to-treat pulmonary pathogen *M. abscessus*, we investigated the feasibility of *M. abscessus* constitutively expressing luciferase from the *luxCDABE* operon as a primary screening strain (*M. abscessus lux*). We showed that expression of *luxCDABE* did not interfere with the growth kinetics of *M. abscessus lux* compared to *M. abscessus* ATCC 19977 by CFU/mL over 72 h, and that *luxCDABE* provided 1,000-fold more luminescence than background (Fig. 1a). More importantly, luminescence production had a positive

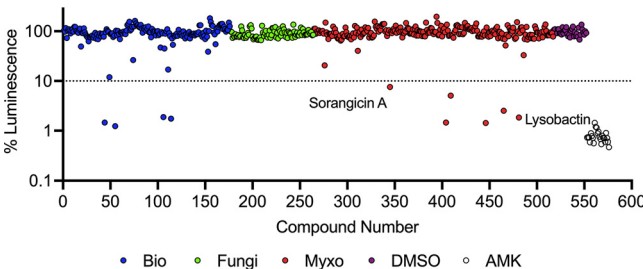

**FIG 2** Natural product primary screen. A library of 517 natural products was screened against *M. abscessus lux* at a concentration of 10 $\mu$M. Dashed red line indicates 90% luminescence reduction cut off. Natural product library includes 176 compounds fractionated from diverse sources (blue, cyan), 88 compounds fractionated from fungi (green), 253 compounds fractionated from myxobacteria (red, orange, yellow), 1% DMSO as vehicle control (purple), and 100 $\mu$M amikacin as positive control (white). Data shown is mean of duplicate screening.

correlation (Pearson $r^2 = 0.9887$) with growth kinetics (Fig. 1b). Therefore, we substituted the luminescence readout as a proxy for bacterial growth.

Primary screens of chemical libraries often include false positives that later prove to be inactive against the microorganism of interest and thus, incur additional time and resources on screening campaigns. Conversely, compounds with bona fide activity may be missed as false negatives. To limit both scenarios, screening conditions are subjected to a Z' factor measurement to determine the robustness of a particular readout. We compared the Z' factor of our *M. abscessus lux* to three other readouts of bacterial growth, namely: optical density at 600 nm ($OD_{600nm}$) (20, 21), resazurin microtiter assay (REMA) (22–24), and Bactiter Glo (25, 26). *M. abscessus* ATCC 19977 and *M. abscessus lux* were grown until mid-log phase, diluted to $OD_{600nm}$ of 0.005 ($5 \times 10^6$ CFU/mL), and incubated with 1% DMSO or 100 $\mu$M amikacin in a 96-well plate format for 48 h. We measured the highest Z' factor of 0.70 (95% CI,0.65 to 0.82) using luminescence from *M. abscessus lux* while luminescence from the commercial Bactiter Glo kit generated the lowest Z' factor of 0.51 (95% CI,0.13 to 0.50) (Fig. 1b). Importantly, the reproducibility between Z' factors was highest with *M. abscessus lux* as measured by the 95% CI range (*M. abscessus lux* = 0.17, REMA = 0.23, Bactiter Glo = 0.38, OD = 0.39). The high Z' factor and its reproducibility for the *M. abscessus lux* readout gave us confidence to use it in our primary screening assays (Fig. 1c).

**Lysobactin and sorangicin A are identified as inhibitors of *M. abscessus* ATCC 19977 from a natural product library.** To identify novel antimicrobials against *M. abscessus*, we carried out a phenotypic screen with an in-house HZI/HIPS library of 517 natural products against *M. abscessus lux*. The natural products originated from the fermentation of various biological sources, fungi, and myxobacteria. Fermentations were fractionated into single compounds at 1 mM in DMSO. We screened the natural products at 10 $\mu$M (1% DMSO) and applied a threshold of 90% and 50% reduction in luminescence compared to drug free controls (1% DMSO). These criteria provided us with compounds with $MIC_{90}$ and $MIC_{50} \leq 10$ $\mu$M, respectively. The last column of each plate contained drug free wells and 100 $\mu$M AMK wells as negative and positive controls, respectively. After screening in duplicate, we identified 12 compounds that met our threshold of 90% loss of viability and 20 compounds at 50% loss of viability at 10 $\mu$M (Fig. 2, Table S1). Many of the compounds identified are known DNA intercalators and were flagged as potentially cytotoxic. Telithromycin was omitted through de-replication as it shares a similar mechanism of action to CLR. Two compounds identified with specific targets were LYB (Fig. 3) and SOR (Fig. 4). LYB was a compound of interest since there are currently no lead compounds that target lipid II in the cell wall in the *M. abscessus* drug pipeline (27). Although traditional cyclic peptides like vancomycin (VAN) do exhibit *in vitro* activity against *M. abscessus* (28), VAN is associated with an increased risk ratio for total adverse events, nephrotoxicity, and vancomycin flushing reaction (29). SOR, a known RNA polymerase (RNAP) inhibitor, was a compound of interest since RNAP is a validated drug target in *Mycobacterium tuberculosis* targeted by the front-line drug rifampicin (RIF) (30). Relatedly, rifabutin (RFB) has been shown to be active against *M. abscessus* (31, 32), and so SOR could be added to the group of repurposed RNAP inhibitors for *M. abscessus*.

Lysobactin

Vancomycin

**FIG 3** Structures of nonribosomal peptide synthetase-derived cyclic peptides targeting peptidoglycan biosynthesis.

**Lysobactin and sorangicin A maintain activity in different media compositions.** To confirm the activity of LYB and SOR and determine the minimum concentration of each drug that inhibits 90% of bacterial growth ($MIC_{90}$), fresh LYB and SOR powder was obtained and tested against the *M. abscessus* ATCC 19977 reference strain. We observed low micromolar dose-response activity for both LYB and SOR as illustrated in Table 1. It was previously shown that some antimycobacterial agents have carbon dependent activities (33). Therefore, we measured the $pMIC_{90}$ of LYB and SOR for *M. abscessus* ATCC 19977 grown in traditional Middlebrook 7H9 with glycerol, Middlebrook 7H9 with acetate, and Sauton's minimal mycobacteria medium with glycerol. Alternatively, from a clinical perspective, the Clinical and Laboratory Standards Institute guidelines recommend *M. abscessus* drug susceptibility testing in cation-adjusted Mueller-Hinton (CaMH) broth (34). LYB and SOR retained activity across glycerol and acetate as carbon sources, in a minimal media, and in media adjusted for $Mg^{2+}$ and $Ca^{2+}$ (Table 1, Table S2).

**Lysobactin and sorangicin A are active against the *M. abscessus* complex and drug resistant clinical isolates.** To compare the activity of LYB and SOR against other drugs with a similar target, we measured the $pMIC_{90}$ of *M. abscessus* ATCC 19977 S and R reference strains against VAN as a representative cyclic peptide cell wall antimicrobial, and RIF and RFB as representative RNAP inhibitors. LYB and VAN demonstrated similar potencies against both *M. abscessus* S and *M. abscessus* R reference strains while SOR and RIF exhibited lower potencies than RFB (Table 2 and 3, Table S3). To determine if LYB and SOR are active against the *M. abscessus* complex, we measured the $pMIC_{90}$ of the natural products against drug resistant clinical isolates that include *M. abscessus* subsp. *abscessus* (*n* = 5), *M. abscessus* subsp. *massiliense* (*n* = 5), and *M. abscessus* subsp. *bolletii* (*n* = 1). These clinical isolates are resistant to a variety of antibiotics used to treat *M. abscessus* pulmonary disease (Table S4). Whole-genome sequencing of the clinical isolates identified SNPs in *erm*(41), *rrl*, and *rpoB* that were confirmed with Sanger sequencing (Table 2). Clinical isolates represented T28 sequevars with a functional *erm*(41) for inducible CLR resistance and C28 sequevars with a nonfunc-

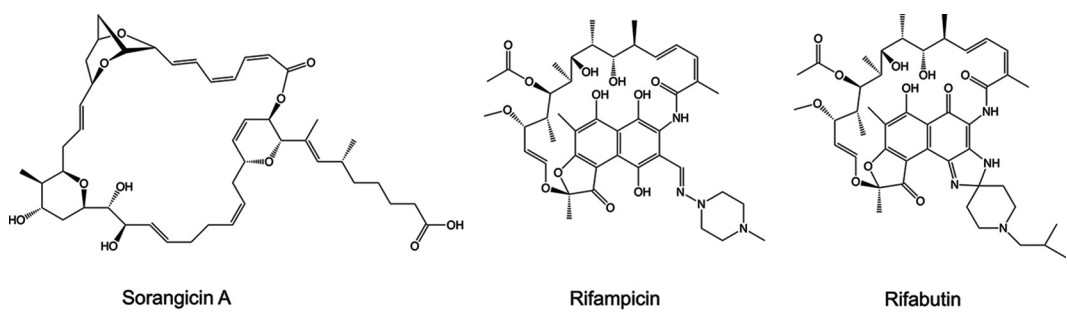

**FIG 4** Structures of polyketide synthase-derived sorangicin A and semisynthetic rifamycins targeting RNA polymerase.

**TABLE 1** Effect of carbon source and minimal media on potencies of natural product hits against *M. abscessus* ATCC 19977 smooth reference strain

| Compound | pMIC$_{90}$ (M) | | | |
|---|---|---|---|---|
| | 7H9$^a$ glycerol | 7H9$^a$ acetate | CaMH$^b$ glycerol | Sauton$^c$ glycerol |
| Lysobactin | 5.5 | 5.5 | 5.4 | 5.3 |
| Sorangicin A | 4.9 | 4.8 | 4.8 | 4.4 |

$^a$Middlebrook 7H9 media.
$^b$Cation-adjusted Mueller Hinton media.
$^c$Sauton mycobacteria minimal media.

tional *erm*(41). Two strains possessed the A2058G (A2270G in *M. abscessus*) or A2059C (A2271C in *M. abscessus*) SNP in the *rrl* gene that conferred constitutive CLR resistance (Table 2). Most isolates harbored a D523E substitution in RpoB. LYB and VAN demonstrated nearly equipotent activities against the clinical isolates (Median LYB pMIC$_{90}$ = 5.3 [95% CI,4.9 to 5.4], VAN pMIC$_{90}$ = 5.3 [95% CI,5.1 to 5.6], *P* = 0.46) (Fig. 5a). SOR and RIF were less potent compared to best-in-class RFB (Median SOR pMIC$_{90}$ = 4.9 [95% CI,4.7 to 5.1], RIF pMIC$_{90}$ = 4.9 [95% CI,4.4 to 5.2], RFB pMIC$_{90}$ = 5.8 [95% CI,5.5 to 5.9]) (Fig. 5b). However, to measure preexisting resistance to the new compounds, pMIC$_{90}$ values were standardized to the ATCC 19977 S reference strain. We measured up to a 10-fold change in pMIC$_{90}$ of the clinical isolates relative to the ATCC 19977 reference strain for LYB with median LYB ΔpMIC$_{90}$ = 0.4 [95% CI,0.3 to 0.9] and VAN ΔpMIC$_{90}$ = −0.1 [95% CI, −0.4 to 0.1] (Fig. 5a). Although the RpoB$^{D523E}$ variant was commonly identified in these *M. abscessus* clinical isolates, *M. tuberculosis* naturally encodes E523 in the rifampicin resistance determining region. Fig. 5b illustrates a similar activity spectrum to SOR, RIF, and RFB. Taken together, these data suggest a lack of acquired resistance in the clinic to LYB and SOR.

## DISCUSSION

Ten hits were identified from a library of 517 natural products from fermented microorganisms. Two compounds of interest are LYB and SOR. LYB, also known as katanosin B, is a cyclic depsipeptide secondary metabolite produced by *Lysobacter ezymogenes* (35, 36). First identified in 1988, LYB was found to inhibit peptidoglycan synthesis in Gram-positive bacteria, but its molecular mechanism remained undefined (35, 36). Later, it was shown that while VAN binds to the terminal d-Ala-d-Ala residue of the pentapeptide stem on N-acetylmuramic acid/*N*-acetylglucosamine units, LYB binds to the reducing end of the lipid-anchored peptidoglycan precursor, lipid II (27). Cyclic peptides have recently been investigated for activity against *M. abscessus*. Teicoplanin showed synergy in combination with the glycylcycline, tigecycline, and VAN synergized with the macrolide, clarithromycin (37, 38). Due to previous work showing that LYB causes moderate toxicity in mice when administered intravenously (36) but the absence of clinical resistance shown here, LYB could benefit from novel formulations to increase the oral bioavailability and limit side effects.

**TABLE 2** Characterization of *M. abscessus* clinical isolates

| Isolate | Subspecies | Morphotype | *rpoB* AA | *rrl* SNP | *erm*(41) sequevar | Clarithromycin susceptibility |
|---|---|---|---|---|---|---|
| ATCC19977 | *abscessus* | Smooth | None | None | T28 | Sensitive |
| ATCC19977 | *abscessus* | Rough | None | None | T28 | Sensitive |
| MB084806 | *abscessus* | Smooth | D523E | None | T28 | Sensitive |
| MB092927 | *abscessus* | Smooth | D523E | None | C28 | Sensitive |
| MB093261 | *abscessus* | Smooth | None | None | T28 | Sensitive |
| L0007906 | *abscessus* | Rough | D523E | A2270G | T28 | Resistant |
| MB086151 | *abscessus* | Rough | D523E | None | C28 | Sensitive |
| MB088425 | *massiliense* | Smooth | D523E | None | Deletion | Sensitive |
| MB088215 | *massiliense* | Smooth | D523E | None | Deletion | Sensitive |
| MB092961 | *massiliense* | Rough | D523E | None | Deletion | Sensitive |
| L00042522 | *massiliense* | Rough | D523E | A2271C | Deletion | Resistant |
| AV | *massiliense* | Smooth | D523E | None | Deletion | Sensitive |
| 167P | *bolletii* | Rough | D523E | None | T28 | Resistant |

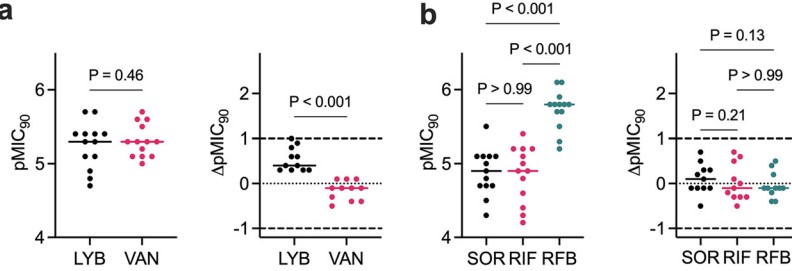

**FIG 5** Potencies of natural product hits against *M. abscessus* ATCC 19977 S and R reference strains and clinical isolates. Clinical isolates comprise *M. abscessus* complex (*M. abscessus* n = 5, *M. massiliense* n = 5, *M. bolletii* n = 1). (a) $MIC_{90}$ ($\mu$M) of LYB and VAN converted to $pMIC_{90}$ (M) (left). Change in $pMIC_{90}$ relative to *M. abscessus* ATCC 19977 ($pMIC_{90\ ATCC} - pMIC_{90\ clin\ iso}$). Values greater than 0 indicate lower potency against the clinical isolate. Dashed lines represent 10-fold change in potency. Data shown is median with *P*-values from Wilcoxon matched-pairs rank test. (b) $MIC_{90}$ ($\mu$M) of SOR, RIF, and RFB converted to $pMIC_{90}$ (M) (left). Change in $pMIC_{90}$ relative to *M. abscessus* ATCC 19977 ($pMIC_{90\ ATCC} - pMIC_{90\ clin\ iso}$). Values greater than 0 indicate lower potency against the clinical isolate. Dashed lines represent 10-fold change in potency. Data shown is median with *P*-values from Friedman test with Dunn's multiple-comparison test.

SOR is produced by the gliding myxobacterium *Sorangium cellulosum* and targets eubacterial but not eukaryotic RNAPs (39, 40). Like RIF and RFB, SOR inhibits bacterial transcription via binding to the RpoB subunit of wild-type RNAP but in *M. tuberculosis*, SOR was shown to prevent promoter DNA unwinding specifically in RIF[R] mutants (30, 41). Importantly, SOR maintained activity against *E. coli* and *M. tuberculosis* *rpoB* mutants despite being resistant to RIF. This has important implications for the treatment of *M. tuberculosis* as RIF is a front-line agent and for the treatment of *M. abscessus* since the activity of RFB has recently been explored (31, 32, 42). SOR could be a promising candidate for the *M. abscessus* drug pipeline as it not only retains activity against *M. tuberculosis* RIF[R] mutants but induces cytochrome P450 3A4 to a lesser degree than the classic RNAP inhibitors RIF and RFB (30). Cytochrome P450 3A4 induction from rifampicin has been shown to reduce the efficacy of the *cftR* corrector, ivacaftor, for patients with CF through drug-drug interactions (43).

*M. abscessus* drug discovery could benefit from additional library screening to identify novel scaffold/drug target pairs like the oxaboroles and tRNA synthetases, but attrition rates are high (44–46). Alternatively, focusing on pharmacologically and clinically validated drug targets like peptidoglycan and RNAP might accelerate the drug discovery process (47). Although inhibitors for these targets do not exert the most potent *in vitro* activity, they provide novel scaffolds with subtly different mechanism of action, whose potency and drug disposition properties can be improved through careful medicinal chemistry optimization or

**TABLE 3** Potencies of natural product hits against *M. abscessus* reference strain and clinical isolates

| Isolate | Subspecies | Morphotype | $pMIC_{90}$ (M)[a] | | | | |
|---|---|---|---|---|---|---|---|
| | | | Cell wall | | RNA polymerase | | |
| | | | LYB | VAN | SOR | RIF | RFB |
| ATCC19977 | *abscessus* | Smooth | 5.7 | 5.2 | 5.0 | 4.9 | 5.7 |
| ATCC19977 | *abscessus* | Rough | 5.7 | 5.0 | 4.7 | 4.6 | 5.7 |
| MB084806 | *abscessus* | Smooth | 5.4 | 5.6 | 5.1 | 5.2 | 6.1 |
| MB092927 | *abscessus* | Smooth | 5.1 | 5.5 | 4.7 | 4.8 | 5.8 |
| MB093261 | *abscessus* | Smooth | 5.1 | 5.1 | 4.8 | 5.0 | 5.8 |
| L0007906 | *abscessus* | Rough | 4.9 | 5.3 | 4.7 | 4.2 | 5.3 |
| MB086151 | *abscessus* | Rough | 5.3 | 5.6 | 4.9 | 4.9 | 5.8 |
| MB088425 | *massiliense* | Smooth | 5.4 | 5.3 | 5.1 | 5.2 | 5.8 |
| MB088215 | *massiliense* | Smooth | 5.4 | 5.3 | 5.1 | 5.1 | 5.8 |
| MB092961 | *massiliense* | Rough | 5.4 | 5.3 | 5.1 | 5.2 | 5.9 |
| L00042522 | *massiliense* | Rough | 4.7 | 5.1 | 4.3 | 4.3 | 5.2 |
| AV | *massiliense* | Smooth | 4.8 | 5.1 | 4.5 | 4.4 | 5.5 |
| 167P | *bolletii* | Rough | 5.3 | 5.7 | 5.5 | 5.4 | 6.1 |

[a]LYB, lysobactin; VAN, vancomycin; SOR, sorangicin A; RIF, rifampicin; RFB, rifabutin.

alternative biotechnological approaches for compound improvement (16). Future studies should include structure activity relationship analysis and novel formulations to improve the activity and bioavailability while minimizing toxicity of the natural products LYB and SOR for the development of *M. abscessus* antibiotics.

## MATERIALS AND METHODS

**Compounds.** Myxobacteria and fungi are prolific sources of structurally diverse metabolites displaying innovative modes-of-action (48, 49). The labs at HZI/HIPS focus on exploring understudied sources and expanding the natural product space in biodiversity-driven approaches (15). Production and isolation procedures are being developed and adapted to match compound properties, and we typically aim at isolating compounds at > 90% purity. The library plates and master stocks are stored at −80°C and undergo routine quality control via HPLC-MS/UV. For hit confirmation and dose response, independent powder stocks are provided which are typically at 95% purity. An in-house collection of 517 natural products were included in the screen. The library includes 176 compounds isolated from diverse sources, 88 compounds isolated from fungi, and 253 compounds isolated from myxobacterial fermentation. The natural products were prepared at 1 mM in 100% dimethyl sulfoxide (DMSO). Amikacin (AMK) was resuspended in distilled $H_2O$ (d$H_2O$). Vancomycin (VAN) was resuspended in d$H_2O$ while lysobactin (LYB), rifampicin (RIF), and rifabutin (RFB) were resuspended in 100% DMSO. Sorangicin A (SOR) was prepared in-house and resuspended in 100% DMSO. For Sorangicin A, fermentation and downstream processing is described elsewhere and the provided sample was > 95% purity. All drugs were purchased from Sigma-Aldrich unless otherwise stated.

**Bacterial strains.** *Mycobacterium abscessus* ATCC 19977 reference strain was used to create a constitutively luminescent strain by integration of the *luxCDABE* operon (*M. abscessus lux*) for screening assays. The integration was accomplished by electroporating the plasmid plux that targets the attP site of the mycobacterial genome and selecting on 7H10 plates containing 250 $\mu$g/mL of kanamycin. *M. abscessus lux* cultures were grown in 100 $\mu$g/mL kanamycin for plasmid maintenance. Cultures were passaged in antibiotic-free media one night prior to the primary screen assay. Clinical isolates of *M. abscessus* subsp. *abscessus* and subsp. *massiliense* from patients with cystic fibrosis were obtained from an epidemiologic study of *M. abscessus* transmission on the island of Montreal. Isolates were characterized by Illumina Mi-Seq whole-genome sequencing. *M. abscessus* subsp. *bolletii* clinical isolate was obtained from France. Single nucleotide polymorphisms (SNPs) in *erm*(41), *rpoB*, and *rrl* were detected using Galaxy. Briefly, whole-genome sequence qualities were checked with FASTQC, and adapter sequences were trimmed with Trimmomatic. Snippy was used to map whole-genome sequences to *erm*(41), *rpoB*, and *rrl* from ATCC 19977 reference (GenBank accession: CU458896.1) with a minimum mapping quality of 60 and minimum coverage of 40. SNPs in *erm*(41), *rpoB*, and *rrl* were confirmed with Sanger sequencing. *erm*(41) primers: $P_F$ GT GTCCGGCCAACGGTCGCGA; $P_R$ TCAGCGCCGCCTGATCACCAGC; *rpoB* primers: $P_F$ TGTCGCAGTTCATGGACCAG AA; $P_R$ GTCGTGCTCGAGGAACGGGAT; *rrl* primers: $P_F$ GACGATGTATACGGACTGACGC; $P_R$ CGTCCAGGTTGAGGG AACCTT. Phenotypic clarithromycin resistance was confirmed genotypically by *erm*(41) sequevar (T28 for active *erm*[41] or C28 for inactive *erm*[41]) or A2058/2059 polymorphisms in *rrl*.

**Culture conditions.** *M. abscessus* strains were grown in rolling liquid culture at 37°C in Middlebrook 7H9 broth (BD Difco) supplemented with 10% (vol/vol) albumin dextrose catalase enrichment (ADC), 0.2% (vol/vol) glycerol, and 0.05% (vol/vol) Tween 80 (7H9 complete) or on 7H10 agar plates supplemented with 10% (vol/vol) oleic acid ADC enrichment and 0.5% (vol/vol) glycerol at 37°C unless otherwise stated; 7H9 broth supplemented with 10% (vol/vol) ADC enrichment, 0.2% (vol/vol) sodium acetate, and 0.05% (vol/vol) Tween 80 was used as an alternative carbon source. Cation-adjusted Mueller-Hinton broth (BD Difco) with 10% (vol/vol) ADC enrichment, 0.2% (vol/vol) glycerol, and 0.05% (vol/vol) Tween 80 was used as an alternative media and bacteria were grown at 30°C. Sauton's minimal medium was used as a defined minimal media (0.5 g/L monobasic potassium phosphate, 4.0 g/L ʟ-asparagine monohydrate, 2.0 g/L citric acid monohydrate, 0.05 g/L ferric ammonium citrate, 0.1 mL of 1% zinc sulfate, 0.5 g/L magnesium sulfate heptahydrate, 60 mL of 100% glycerol, 2.5 mL of 20% Tween 80, pH 7).

**Screening assay.** The natural product library was screened against *M. abscessus lux* at a single-point concentration of 10 $\mu$M in duplicate in 96-well flat-bottom white plates in 7H9 complete media. The culture was grown to log phase (OD$_{600}$ 0.4–0.8) and diluted to an OD$_{600}$ of 0.005 ($5\times10^6$ CFU/mL). 90 $\mu$L of culture was mixed with 10 $\mu$L of 100 $\mu$M compound diluted in a mixture of 10% DMSO/90% 7H9. The final concentration of DMSO in each well was 1%. Plates were sealed with parafilm and incubated at 37°C for 48 h. The last column of each plate had 1% DMSO as negative control in quadruplicate (drug-free conditions) and 110 $\mu$M AMK as positive control in quadruplicate. Luminescence was measured with an Infinite F200 Tecan plate reader. Percentage of luminescence relative to DMSO control was plotted using GraphPad Prism version 9. Compounds that decreased the luminescence to ≤ 10% or ≤ 50% of the drug-free conditions were classified as strong or moderate hits, respectively.

**Determination of MICs.** MIC values were determined using the resazurin microtiter assay (REMA). Cultures were grown to log phase (OD$_{600}$ of 0.4–0.8) and diluted to OD$_{600}$ of 0.005 ($5\times10^6$ CFU/mL). Drugs were prepared in 2-fold serial dilutions in 96-well plates with 90 $\mu$L of bacteria per well to a final volume of 100 $\mu$L. 96-well clear, flat-bottom plates were incubated at 37°C until drug-free wells were turbid (48 h for *M. abscessus*). 10 $\mu$L of resazurin (0.025% wt/vol) was added to each well. Once the drug-free wells turned pink (3 to 4 h), the fluorescence (ex/em 560 nm/590 nm) was measured using an Infinite F200 Tecan plate reader. Fluorescence intensities were converted to % viable cells relative to drug-free conditions and fit to the modified Gompertz equation using GraphPad Prism version 9. MIC values at 90% growth inhibition were determined from the

nonlinear regression Gompertz equation (50). To compare MICs across media or clinical isolates, MIC values were log transformed (pMIC = −log[MIC]).

## SUPPLEMENTAL MATERIAL

Supplemental material is available online only.

**SUPPLEMENTAL FILE 1**, PDF file, 0.3 MB.

## ACKNOWLEDGMENT

The *luxCDABE* plasmid was kindly gifted by Jeffery S. Cox.

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
