## [Reviewer comments · Microbiology Spectrum]

Microbiology Spectrum

Natural products lysobactin and sorangicin A show *in vitro* activity against *Mycobacterium abscessus* complex

Jaryd Sullivan, Jacqueline Yao, Christophe Courtine, Andréanne Lupien, Jennifer Herrmann, Rolf Müller, and Marcel Behr

Corresponding Author(s): Marcel Behr, McGill University Health Centre

Review Timeline:

Submission Date:	July 12, 2022
Editorial Decision:	August 24, 2022
Revision Received:	September 16, 2022
Accepted:	October 8, 2022

Editor: Amit Singh

Reviewer(s): Disclosure of reviewer identity is with reference to reviewer comments included in decision letter(s). The following individuals involved in review of your submission have agreed to reveal their identity: Gerard D Wright (Reviewer #3)

Transaction Report:

DOI: <https://doi.org/10.1128/spectrum.02672-22>

August 24, 2022

Dr. Marcel A Behr
McGill University Health Centre
1001 boul Décarie
Glen Site Block E, Office #E05.1608
Montreal H4A 3J1
Canada

Re: Spectrum02672-22 (Natural products lysobactin and sorangicin A show *in vitro* activity against *Mycobacterium abscessus* complex)

Dear Dr. Marcel A Behr:

Our reviewers are in agreement that your work is very important for the field. However, reviewer #1 and 2 have raised some minor concerns for you to consider. Kindly address these concerns appropriately in a revised manuscript.

Link Not Available

Sincerely,

Amit Singh

Journals Department
Reviewer comments:

Reviewer #1 (Comments for the Author):

Behr and co-workers screened an in-house unique natural product library of the Helmholtz-Institute derived from a rare collection of myxobacteria and fungi against *Mycobacterium abscessus* (Mab) using a lux-reporter strain and identified two natural products lysobactin and sorangicin with useful activity through inhibition of peptidoglycan biosynthesis and RNA polymerase, respectively. The authors further characterized the activities of these two natural products in different media and Mab clinical isolates. The overall scholarship is excellent and the work was rigorously performed. *M. abscessus* is notoriously drug-resistant,

so the identification of novel scaffold against established targets is deemed highly significant. Sorangin inhibits RNA polymerase like the rifamycins and is unaffected by rpoB mutations that confer resistance to the rifamycins. Lysobactin is also quite interesting since its mechanism of action is distinct from glycopeptide antibiotics through binding to the lipid anchor of Lipid II instead of the D-Ala-D-Ala termini of the pentapeptide stem. The authors should consider the following comments in submitting a revision.

Comments.

1. Page 3, line 42 (Introduction first sentence). *M. abscessus* does not generally cause disease in immunocompetent hosts, thus revise the statement "...causes disease in immunocompromised and immunocompetent hosts."
2. Page 3, line 50. Consider editing "Therefore, current treatment guidelines are based empirically on drug susceptibility testing" since empiric treatment typically means based on established susceptibility of given antibiotics. Once susceptibility of a given isolate is determined, then one uses definitive therapy.
3. Page 4, line 65. Consider editing "Natural products are a reservoir of genetically encoded microbial metabolites produced by multimodular biosynthetic gene clusters" since not all natural product biosynthetic pathways are modular (certainly PKS, FAS and NRPSs are modular), but many terpene pathways are not modular, non-ribosomal peptides, glycosides, etc....
4. Page 9, line 168. Vancomycin more precisely binds to the terminal D-Ala-D-ala residue of the pentapeptide stem and not to the terminal pentapeptide.
5. Page 10, lines 190-191. Change "they provide excellent building blocks for iterative medicinal chemistry optimization" to "they provide novel scaffolds with subtly different mechanism of action, whose potency and drug disposition properties can be improved through careful medicinal chemistry optimization."
6. Can the authors provide information on the composition of the library. The negative hits are also valuable data.
7. Information on compound purity should ideally be provided since compounds can degrade.
8. No information is provided on the construction or characterization of the *M. abscessus* lux strain.
9. Screening assay. "90 μ L of culture was mixed with 10 μ L of compound". The authors should specify the solution composition (DMSO or 10% in buffer, etc....). From the experimental, it reads as if 10 μ L of the compound in DMSO was added yielding a final DMSO concentration of 10%. Mycobacteria typically do not grow well above 0.5% DMSO.
10. The cutoff of 90% growth inhibition at 10 μ M may be too strict. A cutoff of >50% growth inhibition would yield a few more hits that could potentially be further optimized.
11. Chemical structure of Vancomycin does not properly display the chirality around the biaryl and biarylether linkages. The stereochemistry of the rifampicin and rifabutin is incorrect at C-23 (should be S, drawn as R) and C-24 (should be R, drawn as S).

Reviewer #2 (Comments for the Author):

The authors have carried out inhibitor identification study against *M. abscessus* since the control measures for *M. abscessus* infections are limited and new drugs are needed for resistant strains. They have screened a natural product library and have found two inhibitors. The manuscript is well written and is an important piece of work. Authors are requested to provide their response to the following:

1. Page 5, line 92: The authors mention O.D. 0.005 as 5×10^6 CFU. Have they determined CFU for this? The authors should comment in the manuscript as to how did they arrive at this CFU or give data for the same?
2. The authors should clearly describe the rationale of the microorganisms selected for the preparation of the library along with the detailed fractionation methods. They should elaborate on how the molecules were separated, their structures determined, or their identification and purity range of the molecules.
3. The authors should elaborate on whether there are any studies on cytotoxicity of the two identified compounds in mammalian cell lines or any other way to determine the same.

Reviewer #3 (Comments for the Author):

The manuscript describes the validation of a luxCDABE engineered strain of *Mycobacterium abscessus* suitable for HTS and its use in a screen of a library of microbial natural products for compounds with antimicrobial activity. The authors identify and confirm the activity of two candidates, the RNA poly inhibitor sorangicin, and the lipid II binder lysobactin. The authors confirm activity vs a panel of clinical isolates, including drug resistant strains.

The paper is well written, the methodology sound, and the discovery of compounds very useful to the field. My only suggestion is that the authors refrain from comments such as " we identified lysobactin and sorangicin A as promising 37 candidates for patients with *M. abscessus* lung infections". The term candidate suggests that these compounds have advanced significantly toward clinical trials. Much work will need to be done before this claim can be used.

Staff Comments:

Preparing Revision Guidelines

Please return the manuscript within 60 days; if you cannot complete the modification within this time period, please contact me. If you do not wish to modify the manuscript and prefer to submit it to another journal, please notify me of your decision immediately so that the manuscript may be formally withdrawn from consideration by Microbiology Spectrum.

Reviewer comment in grey. Our response in black. Text changes in red.

Reviewer #1 (Comments for the Author):

Behr and co-workers screened an in-house unique natural product library of the Helmholtz-Institute derived from a rare collection of myxobacteria and fungi against Mycobacterium abscessus (Mab) using a lux-reporter strain and identified two natural products lysobactin and sorangicin with useful activity through inhibition of peptidoglycan biosynthesis and RNA polymerase, respectively. The authors further characterized the activities of these two natural products in different media and Mab clinical isolates. The overall scholarship is excellent and the work was rigorously performed. M. abscessus is notoriously drug-resistant, so the identification of novel scaffold against established targets is deemed highly significant. Sorangin inhibits RNA polymerase like the rifamycins and is unaffected by rpoB mutations that confer resistance to the rifamycins. Lysobactin is also quite interesting since its mechanism of action is distinct from glycopeptide antibiotics through binding to the lipid anchor of Lipid II instead of the D-Ala-D-Ala termini of the pentapeptide stem. The authors should consider the following comments in submitting a revision.

Thank you for the positive overall feedback and the specific suggestions which are addressed in turn below.

1. Page 3, line 42 (Introduction first sentence). M. abscessus does not generally cause disease in immunocompetent hosts, thus revise the statement "...causes disease in immunocompromised and immunocompetent hosts."

We included both immunocompromised and immunocompetent hosts as to not omit the patients that acquire M. abscessus skin infections after plastic surgeries, or other surgical interventions. Two references to surgical wound infections were added.

2. Page 3, line 50. Consider editing "Therefore, current treatment guidelines are based empirically on drug susceptibility testing" since empiric treatment typically means based on established susceptibility of given antibiotics. Once susceptibility of a given isolate is determined, then one uses definitive therapy.

We have removed the sentence, because it only added confusion to the paragraph.

3. Page 4, line 65. Consider editing "Natural products are a reservoir of genetically encoded microbial metabolites produced by multimodular biosynthetic gene clusters" since not all natural product biosynthetic pathways are modular (certainly PKS, FAS and NRPSs are modular), but many terpene pathways are not modular, non-ribosomal peptides, glycosides, etc....

Line about gene clusters was omitted. Replaced with this line and added two supporting references on page 4, line 68.

"Natural products are a reservoir of genetically encoded microbial metabolites with vast chemical diversity."^{15,16,}

4. Page 9, line 168. Vancomycin more precisely binds to the terminal D-Ala-D-ala residue of the pentapeptide stem and not to the terminal pentapeptide.

Clarification noted and changed accordingly on page 9, line 176.

"Later, it was shown that while VAN binds to the terminal D-Ala-D-Ala residue of the pentapeptide stem on N-acetylmuramic acid/N-acetylglucosamine units, LYB binds to the reducing end of the lipid-anchored peptidoglycan precursor, lipid II."^{28,}

5. Page 10, lines 190-191. Change "they provide excellent building blocks for iterative medicinal chemistry optimization" to "they provide novel scaffolds with subtly different mechanism of action, whose potency and drug disposition properties can be improved through careful medicinal chemistry optimization."

Thank you for the improved line. It now reads on page 10 line 201:

“Although inhibitors for these targets do not exert the most potent *in vitro* activity, they provide novel scaffolds with subtly different mechanism of action, whose potency and drug disposition properties can be improved through careful medicinal chemistry optimization or alternative biotechnological approaches for compound improvement.¹⁹”

6. Can the authors provide information on the composition of the library. The negative hits are also valuable data.

The library consists of isolated natural products from myxobacteria, fungi, streptomycetes and few other sources. These compound display antibacterial, antiviral, antifungal, antiparasitic, cytotoxic, and immunomodulatory properties. The plates are part of the DZIF natural product collection. (<https://www.dzif.de/en/compound-resources-and-medicinal-chemistry>)

7. Information on compound purity should ideally be provided since compounds can degrade.

The Methods section on page 11 line 215 the following lines were added for clarification:

“The library plates and master stocks are stored at -80°C and undergo routine quality control via HPLC-MS/UV. For hit confirmation and dose-response, independent powder stocks are provided which are typically at 95% purity.”

8. No information is provided on the construction or characterization of the *M. abscessus* lux strain.

The characterization of the lux strain was in Figure 1, but we did not provide experimental details on its construction. In the Methods, page 11, line 229, we have added the following text:

“The integration was accomplished by electroporating the plasmid *plux* that targets the *attP* site of the mycobacterial genome and selecting on 7H10 plates containing 250 µg/mL of kanamycin.”

9. Screening assay. "90 µL of culture was mixed with 10 µL of compound". The authors should specify the solution composition (DMSO or 10% in buffer, etc...). From the experimental, it reads as if 10 µL of the compound in DMSO was added yielding a final DMSO concentration of 10%. Mycobacteria typically do not grow well above 0.5% DMSO.

Agree that the sentence is confusing. It has been modified accordingly on page 13, line 266:

“90 µL of culture was mixed with 10 µL of 100 µM compound diluted in a mixture of 10% DMSO/90% 7H9. The final concentration of DMSO in each well was 1%.”

10. The cutoff of 90% growth inhibition at 10 µM may be too strict. A cutoff of >50% growth inhibition would yield a few more hits that could potentially be further optimized.

The 90% cut-off was chosen arbitrarily. The supplementary table with hits will be updated to include both cut-offs.

11. Chemical structure of Vancomycin does not properly display the chirality around the biaryl and biarylether linkages. The stereochemistry of the rifampicin and rifabutin is incorrect at C-23 (should be S, drawn as R) and C-24 (should be R, drawn as S).

Thank you for the corrections. Structures have been modified accordingly.

Reviewer #2 (Comments for the Author):

The authors have carried out inhibitor identification study against *M. abscessus* since the control measures for *M. abscessus* infections are limited and new drugs are needed for resistant strains. They have screened a natural product

library and have found two inhibitors. The manuscript is well written and is an important piece of work. Authors are requested to provide their response to the following:

1. Page 5, line 92: The authors mention O.D. 0.005 as 5×10^6 CFU. Have they determined CFU for this? The authors should comment in the manuscript as to how did they arrive at this CFU or give data for the same?

The OD_{600nm} of a liquid culture of *M. abscessus* was measured and serially diluted 10-fold. 100uL aliquots of the dilutions were plated on 7H10 agar. The dilution with countable CFU was used to calculate CFU/mL in the original culture. The ratio of OD 1 $\sim 1 \times 10^9$ CFU/mL is used for *M. abscessus* work in the Behr lab. When an accurate CFU/mL count is needed, cultures are serially diluted for CFU counts for that experiment.

2. The authors should clearly describe the rationale of the microorganisms selected for the preparation of the library along with the detailed fractionation methods. They should elaborate on how the molecules were separated, their structures determined, or their identification and purity range of the molecules.

The Methods section on page 11 line 211 and 224 the following lines were added for clarification:

“Myxobacteria and fungi are prolific sources of structurally diverse metabolites displaying innovative modes-of-action.^{49,50} The labs at HZI/HIPS focus on exploring understudied sources and expanding the natural product space in biodiversity-driven approaches.¹⁵ Production and isolation procedures are being developed and adapted to match compound properties, and we typically aim at isolating compounds at > 90% purity.”

“For Sorangicin A, fermentation and downstream processing is described elsewhere and the provided sample was > 95% purity.”

3. The authors should elaborate on whether there are any studies on cytotoxicity of the two identified compounds in mammalian cell lines or any other way to determine the same.

Toxicology is being assessed with each iterative medicinal chemistry cycle and with formulations.

Reviewer #3 (Comments for the Author):

The manuscript describes the validation of a luxCDABE engineered strain of *Mycobacterium abscessus* suitable for HTS and its use in a screen of a library of microbial natural products for compounds with antimicrobial activity. The authors identify and confirm the activity of two candidates, the RNA poly inhibitor sorangicin, and the lipid II binder lysobactin. The authors confirm activity vs a panel of clinical isolates, including drug resistant strains. The paper is well written, the methodology sound, and the discovery of compounds very useful to the field. My only suggestion is that the authors refrain from comments such as " we identified lysobactin and sorangicin A as promising 37 candidates for patients with *M. abscessus* lung infections". The term candidate suggests that these compounds have advanced significantly toward clinical trials. Much work will need to be done before this claim can be used.

Thank you for the suggestion. We modified the sentence accordingly on page 2, line 36:

“Using an in-house library of curated natural products, we identified lysobactin and sorangicin A as novel scaffolds for the future development of antimicrobials for patients with *M. abscessus* infections.”

October 8, 2022

Dr. Marcel A Behr
McGill University Health Centre
1001 boul Décarie
Glen Site Block E, Office #E05.1608
Montreal H4A 3J1
Canada

Re: Spectrum02672-22R1 (Natural products lysobactin and sorangicin A show *in vitro* activity against *Mycobacterium abscessus* complex)

Dear Dr. Marcel A Behr:

Your manuscript has been accepted, and I am forwarding it to the ASM Journals Department for publication. You will be notified when your proofs are ready to be viewed.

Sincerely,

Amit Singh
Editor, Microbiology Spectrum
